# Forecasting the Mediterranean Sea Marine Heatwave of summer 2022

Ronan McAdam[1], Giulia Bonino[1], Emanuela Clementi[1], Simona Masina[1]

[1]CMCC Foundation – Euro-Mediterranean Center on Climate Change, Italy

*Correspondence to*: Ronan McAdam (ronan.mcadam@cmcc.it)

**Abstract.** Early-warning of marine heatwaves requires short-term forecasts to provide precise information on timings, local-scale coverage and intensities of coming events. Here, we describe our successful efforts to track the onset, peak and decay of the Mediterranean Sea marine heatwave of summer 2022 with the Copernicus MedFS short-term (10-day) forecast system. First, we show that the 2022 event eclipses the economically and ecologically damaging event of 2003 in terms of MHW activity (a measure of intensity and duration). Forecasts of MHW area and activity provide a means of basin-wide validation, highlighting the capability of MedFS to capture regional behaviour. On local scales, we found that the MHW occurrence in the Ligurian Sea and Gulf of Taranto, two regions of economic and ecological importance, was also reliably forecast. Encouragingly, we note that the forecast has demonstrated skill in capturing not just the season-long MHW cycle but also breaks in MHW persistence and abrupt changes in local activity. Subseasonal forecasts do not yet demonstrate the capacity to predict MHW response to short-lived weather patterns, but this study confirms that short-term forecasts, at least in the Mediterranean Sea, can fill this gap.

## Short Summary

In the summer of 2022, a regional short-term forecasting system was able to predict the onset, spread, peaks and decay of a record-breaking marine heatwave in the Mediterranean Sea, up to 10 days in advance. Satellite data shows that the event was record-breaking in terms of basin-wide intensity and duration. This study demonstrates the potential of state-of-the-art forecasting systems to provide early-warning of marine heatwaves to marine activities (e.g. conservation and aquaculture).

## 1 Introduction

Disease outbreaks, mass mortality events and the redistribution of species induced by marine heatwaves (MHWs) lead to economic losses to fisheries and aquaculture farms, and hamper conservation efforts (Smith et al., 2021; Garrabou et al., 2022; Smith et al., 2022). The need to prepare for and mitigate these MHW-induced impacts has driven developments in understanding the drivers and predictability of MHWs (Holbrook et al., 2019; Rodrigues et al., 2019; Sen Gupta et al., 2020; Li et al., 2020; Schlegel et al., 2021; McAdam et al., 2023), and in quantifying the skill of forecasts of MHWs (Benthuysen et

al., 2021; Jacox et al., 2022; McAdam et al., 2023). The Mediterranean Sea is a particular "hot-spot" for MHWs, with much literature documenting the increases in intensity, duration, frequency and impacts (Darmaraki et al., 2019; Ibrahim et al., 2019; Juza et al., 2022; Dayan et al., 2023). Despite this, there is currently less information on forecasting capability and event predictability in the Mediterranean region than in others (e.g. the North Pacific: Jacox et al., 2019; de Boisesson et al., 2022). With marine services projected to play an ever-increasing role in global sustainability and economic security (Rayner et al., 2019), early-warning systems of heat extremes can aid their planning and day-to-day management (Hartog et al., 2023).

While inter-annual variability of MHW occurrence and characteristics is derived from ocean warming and preconditioning (de Boisesson et al., 2022), ENSO (Jacox et al., 2022) and atmospheric teleconnections (e.g. Rossby wave trains, Rodrigues et al., 2019), short-lived atmospheric processes and weather systems can disrupt MHW persistence or halt their continuation completely (Benthuysen et al, 2021). The definition of MHWs assumes persistent conditions are harmful to marine life if the duration is 5 days or longer (Hobday et al., 2016), although this number is quite arbitrary and in principle should be species-dependent. The average duration of MHWs across most of the global ocean, as well as in the Mediterranean Sea, falls within the definition of short-term forecasting (< 2 weeks) (Oliver et al., 2021; Dayan et al., 2023). A short-term view of MHWs is therefore crucial to understanding their predictability and their impacts.

Short-term forecasting of MHWs has a range of potential roles in marine activities. While some contingency plans for extreme heat events in the aquaculture and fishing industries require several months notice (e.g. relocating or switching species), others should be performed at the latest possible moment in order to avoid minimise losses (e.g. early harvesting, or cooling of farm water) (Holsman et al., 2019; Galappaththi et al., 2020). In these cases, accurate information on daily timescales is crucial. Short-term forecasts are also useful for marine protected areas (MPAs), allowing them to prepare to monitor ecosystem damage (e.g. coral bleaching) and recovery, which in turn helps assess the effectiveness of their conservation efforts (McLeod et al., 2008). Forecasts of SST can also, in theory, be coupled to distribution models to forecast changes in species habitat for highly-mobile species (Abrahms et al., 2019). "Early-warnings" are a key means of climate resilience for marine services (Galappaththi et al., 2020); an assessment of their ability to track MHWs will contribute to further uptake by these services and unlock potential socio-economic benefits.

During the summer of 2022, the Copernicus Marine Service Mediterranean Physical Forecasting system (MedFS) was employed to monitor and forecast sea surface temperature (SST) increases which eventually evolved into a record-breaking MHW for the region. MedFS has already demonstrated skill in detecting past extreme events in the Mediterranean Sea: the "aqua alta" flooding in Venice in 2019 (Giesen et al., 2020), and Medicane Ianos (Clementi et al., 2022) and Storm Gloria (Alvarez-Fanjul et al., 2022) in 2020, providing evidence that the forecast system has ability to capture a wide range of concurrent conditions (e.g. high surface air temperatures, moisture, atmospheric instability for medicanes; Cavicchia et al, 2014). Here, we provide a basin-wide description of the event and demonstrate the ability of MedFS to accurately predict many

facets of the event (e.g. the onset, spread, persistence and decay). First, we introduce the high-resolution regional forecast
system and the satellite-derived SST data used to identify MHWs. Then, the record-breaking characteristics (intensity,
geographic extent) of the 2022 event are described. We demonstrate the system's ability to predict the MHW spread across the
basin and daily temperature variability in regions of key economic and ecological importance. Finally, we explore the potential
role of short-term forecasting in the early-warning of MHWs compared to other forecasting time scales.

## 2. Dataset & Methods

Here, MHWs are detected with a 0.05° resolution reprocessing of a blend of satellite-derived products provided by the ESA
Climate Change Initiative (CCI) and the Copernicus Climate Change Service (C3S) initiatives, including AVHRR Pathfinder
dataset version 5.3 to increase the input observation coverage (Product ref. no. 1). The dataset provides daily SST of the
Mediterranean Sea from January 1st 1982 to present (currently, up to six months before real time).

The Mediterranean Near Real Time Analysis and Forecast is a 3D coupled hydrodynamic-wave modelling system
implemented at 1/24° (~4 km) horizontal spatial resolution, which produces analysis and 10-day forecasts of the main ocean
essential variables (Product ref. no. 2). The analysis system assimilates satellite sea level anomalies and in-situ temperature
and salinity observations, and nudges SST towards an ultra-high resolution satellite product. The same model framework is
used to provide a multi-decadal reanalysis of the ocean, extending from 1987 to the present (Product ref. no. 3). Forecasts are
made daily; once a week (on Tuesdays) an analysis is used to initialise forecasts, while on other days a hindcast is used. A
schematic of the provision of forecast and analysis data is found in the QUID/PUM of the product.

Forecasts of 2m temperature (T2M) and wind speed are obtained from the European Centre for Medium-Range Weather
Forecasts (ECMWF) operational forecast and analysis distributed by the Italian National Meteorological Service
(USAM/CNMA). Variables are available at 1 hourly resolution for the first 3 days of forecast, 3 hours for the following 3 days
and 6 hourly for the following 4 days. The horizontal resolution is 0.1o. The T2M anomaly is calculated using the same
variables in the ERA5 reanalysis, over the period 1987-2021.

MHWs are defined as SSTs which persist above the 90th-percentile for 5-days or longer (Hobday et al., 2016). Here, the 90th-
percentile threshold corresponds to the 33 year baseline period 1987-2021 calculated individually for satellite derived and
model-derived data, and smoothed with an 11-day moving window. MHWs in MedFS are defined relative to the climatology
of the physical reanalysis. Although there are benefits of detrending SST prior to detecting MHWs (Amaya et al., 2023), we
chose not to detrend in order to present the true values of temperature because they are of more relevance to species impacts
(e.g. Galli et al., 2017).

We use the MHW activity as a means to describe the event on a basin-scale, and to validate the forecast ability to capture the spatial scale of the event. Previously, activity has been defined as the product of event intensity, duration and area over a target period (Simon et al., 2022). Here, in order to study basin-wide spread at daily resolution, we define activity as the sum of the intensity over the area undergoing a MHW in the Mediterranean Basin. We assume that all MHW activity in the basin corresponds to the same event, unlike more novel methods of MHW tracking which employ spatial clustering (Bonino et al, 2023). Nonetheless, the activity metric used here identifies very similar phases of MHW activity during the 2003 event as the more advanced clustering method (Fig 1a; Bonino et al., 2023).

**Product Table**

| Product ref. no | Product ID & type | Data Access | Documentation |
|---|---|---|---|
| 1 | SST_MED_SST_L4_REP_OBSERVATIONS_010_021; Satellite observations | EU Copernicus Marine Service Product (2022a) | PUM: Pisano et al. (2022a) QUID: Pisano et al. (2022b) |
| 2 | MEDSEA_ANALYSISFORECAST_PHY_006_013; Numerical models | EU Copernicus Marine Service Product (2022c) | PUM: Lecci et al. (2022b) QUID: Goglio et al. (2022) |
| 3 | MED_MULTIYEAR_PHYS_006_004; Numerical models | EU Copernicus Marine Service Product (2022b) | PUM: Lecci et al. (2022a) QUID: Escudier et al. (2022) |
| 4 | ECMWF Integrated Forecast System (IFS) Forecast & Analysis | Copernicus Climate Changes Service | https://www.ecmwf.int/en/forecasts/datasets/set-i#I-i-a_fc (Accessed on 13-02-2022) |
| 5 | ERA5 | Copernicus Climate Changes Service | Hersbach et al. (2023) |

## 3. Results

First, we describe the event on a basin scale using satellite observations. We remind the reader that references to specific dates are for indication only, as the precise timings of peaks and onsets may differ when using different datasets and climatologies. In 2022, the onset of summer MHW conditions began in mid-May; by 23rd May, 35% of the area of the Mediterranean Sea was already experiencing MHW conditions (Fig. 1c). Maps of temperature anomaly confirm that the onset occurred mostly in the western regions and the Adriatic Sea (Fig. 2a). The geographic extent of MHW extended into the central and eastern parts of the basin (e.g. Fig. 2b), and MHW area remained above a third of the total basin area until the decay at the end of September. Peak area (70%) was reached on June 6th, while notable peaks of activity occurred later, on June 29th, July 6th and July 27th (Fig. 1a). The peak temperature anomaly of 6.45 °C (above the 1987-2019 average) was reached in the Gulf of Lion on July 18th.

Prior to 2022, the MHWs of the summers of 2003, 2015 and 2018 had been found to have the highest activities on record (using a slightly different definition of activity, but which is still based on intensity and duration; Simon et al., 2022). Here, we find that the activity in 2022 clearly eclipses that of 2015 and 2018, in terms of both maxima and persistence of activity (Fig. 1a). Though the summer of 2003 reached similarly high peaks of activity (twice, in mid-June and at the end of August), the total activity during the summer (defined as the area under the curve) is lower ($82 \times 10^6$ °C.km2) than for 2022 ($139 \times 10^6$ °C.km$^2$). While in 2003 the MHW activity returned to zero in late May and mid-July, in 2022 it persisted throughout the summer above at least 0.5 °C.km2 each day. Using the total activity as a measure, the summer of 2022 now holds the record for MHW activity.

Using the MHW activity provides an efficient, if not complete, means of validating MedFS on the basin-scale. It is important to remember that activity time series cannot identify where and when MHWs are occurring (we study forecast ability in specific regions later). Here, we show both the activity (Fig. 1b) and the area (Fig. 1c) predicted, to infer whether forecast inaccuracies are caused by an inability to capture the geographic extent or the temperature intensity. Overall, we find that MedFS was able to forecast the evolution of basin-wide MHW activity (Fig. 1b). In particular, we highlight the accurate predictions of the timings of the May onset, the various peaks throughout the summer, the two stages of the decay and the September rebound. On several occasions, MHW activity rapidly increases, often doubling or tripling over the period of less than a week; such increases in activity are predicted by the forecasts in mid-May, early-June, mid-June (twice) and mid-July. Timings of declines in activity are also correctly forecast on all occasions, in some cases even with lead times of 5 days or more (e.g., early June). Similar capabilities are found for the forecasts of area of MHW conditions. Ability to capture both the activity and the area

implies accuracy in capturing the intensity as well, although this analysis does not yet determine the geographic distribution
of MHW intensity.

There are indeed forecast inaccuracies to highlight. Firstly, there are instances of peaks of activity being overestimated (e.g.,
by up to a third in early June) and of false alarms about growth being raised (e.g. early August; Fig. 1b). Overestimations of
both activity and area occur throughout the summer, except for the onset in May. Then, there are instances in which MHW
area tendencies follow the activity but are overestimated (late July to early August); given that activity continues to increase
while area decreases (e.g. early September), there is an implied overestimation of the sea surface temperatures. Elsewhere, we
see fluctuations in activity on daily timescales which dominate over the longer-term growth tendencies (e.g., during the growth
period beginning at the end of June). Throughout the summer of 2022, we see various examples of the activity forecasts being
unable to detect this higher-frequency variability. Variability in area, on the other hand, is lower and the forecasts generally
follow the observed tendencies. In summary, the forecasts sometimes persist or increase temperature anomalies for too long,
suggesting that they fail to capture sporadic cooling. Candidate drivers for short term cooling mechanisms not captured by the
forecasts include cloud cover changes or winds.

It is important also to consider the ability to capture the spread of MHW occurrence. The geographical distribution of intensity
in key phases of the MHW life cycle in forecasts agrees well with observations (Fig. 2). During the onset, forecasts capture
the basin-wide patterns, with MHW occurrence at this stage correctly forecast in the Tyrrhenian Sea, Gulf of Lion and parts
of the Adriatic Sea. The spread of the MHW conditions during the peak was correctly predicted to cover the south part of the
Alboran Sea, the Ionian and southern regions of the Levantine Basin. Meanwhile, the Aegean Sea was predicted to be shielded
from MHWs and instead experience cold anomalies, most likely caused by cooling related to the Etesian winds (Poupkou et
al., 2011). Lastly, the first decay phase at the end of August produced very inhomogeneous MHW conditions across the basin.
This "patchiness", indicative of local-scale processes acting to cool the ocean such as increased cloud cover or weak winds,
was indeed predicted, but how well the forecast matches observations depends greatly on the local regions of interest and the
exact day and lead time considered. Although it is not possible to draw rigorous conclusions from snapshots, the accuracy of
basin-wide occurrence (Fig. 1b) suggests that forecast ability to capture MHW patterns and spread was generally high across
the entire summer.

While basin-scale analysis allows an overview of forecasting skill, local-scale testing is imperative as forecasting tools are expected to be used on local-scale analysis (Dayan et al., 2023). Here, we also provide MHW forecasts for two key areas of maritime activity in the Mediterranean Sea: the Ligurian Sea and the Gulf of Taranto (Fig. 3). Each region experienced MHW conditions at different times during the summer, and in each case the forecasts accurately predicted the onset, persistence, intensity and decays. The Ligurian Sea, bordered by Italy and France, is a crucial location for marine conservation; it doubles as a marine protected area (the Pelagos Sanctuary for Mediterranean Marine Mammals) which is home to unique species of fin whales and striped dolphins, amongst other species (Notarbartolo-de-Sciara et al., 2008). The Ligurian Sea experienced 115 days of MHWs throughout the summer, and temperature anomalies reached a maximum of 4.46 ºC above the 1987-2019 average at the end of July, coinciding with the peak temperature of the summer (28.74 ºC). The forecasts of SST were highly accurate; root-means-squared difference (RMSE) of the forecasts shown in Fig. 3a (across lead times) was 0.12º C, lower than the reported RMSDs averaged over the entire Mediterranean Sea between the analysis and independent observations (Goglio et al., 2023). For an indication of forecast reliability, we highlight the false alarms (MHW days forecast but not observed) and misses (MHW days observed but not forecast). First we note that in the 89% of days the correct conditions are forecast with few, sporadic exceptions. For example, the forecast made on May 10th captured the sharp rise in SST but not the MHW conditions at the end of the week. However, reducing the lead time (i.e. checking forecasts made on the 12th or 13th) correctly forecast the MHW state.

The Gulf of Taranto, situated in the Northern Ionian Sea, is one of the most productive areas of shellfish (mussels) farming in Italy (Prioli, 2004) but there is not yet data on MHW-induced mass-mortality or economic loss in this region (Garrabou et al., 2019). Unlike the Ligurian Sea, the Gulf of Taranto experienced three short but intense periods of MHW occurrence in June and July, adding up to 61 days of MHWs in total. The peak temperature anomaly was 4.76 ºC on 6th June, though peak temperatures occurred later in the season. As in the Ligurian Sea, the forecasts were highly accurate (demonstrating a RMSD of 0.08 ºC). Regarding the reliability of MHW forecasts, the continuation of the start of the heatwave in early May was missed by the forecast of May 17th, while the forecast of 2nd August missed several days of MHW occurence. However, in both cases, SST increases were predicted. None of the forecasts shown in Figure 3 raised false alarms.

So far we have studied accuracy of the entire forecast period but, in some applications, it might be necessary or of more interest to have a specific warning time (e.g. 4 days). As MedFS produces forecasts every day, we now study forecast accuracy for the summer of 2022 at different lead times; we focus on the Western Mediterranean Sea, as opposed to the entire basin, in order to investigate local forcings as reasons for poor skill (Fig. 4). Table 1 quantifies the error of forecasts over the Western Mediterranean Sea. The overestimation of MHW activity in July and August occurred in forecasts with lead time of one day. In many instances, lead time 1 and lead time 4 are similarly far from the observed values, while lead time 7 further

overestimates the peaks in activity (Table 1). MHW area, on the other hand, while predicted more accurately than activity on all lead times, is typically underestimated by the forecasts. This implies an overestimation of SST (i.e. MHW intensity) during the activity peaks. The RMSD normalised by the standard deviation, indicates where errors fall within the range of natural variability (normalised RMSD < 1); in all lead times, this is true for both area and activity, suggesting their skill is similar to atmospheric variables (Table 1).

The decreases in skill with lead time can partly be explained by the decrease in skill of the ECMWF atmospheric forecasts used to force MedFS. T2M and wind speed correlate strongly and significantly with the MHW activity (correlation values of 0.89 and 0.50 with the ECWMF analysis respectively), evidencing their role in MHW formation. Errors of forecasts of T2M and wind speed grow with time but do not exceed natural variability at lead time 7 (Table 1). In the first half of the summer, forecasts at lead time 7 of both T2M and wind speed are frequently out of phase with the observed changes. In fact, the underestimations of MHW area in this period occur simultaneously with underestimations of T2M. For example, the underestimation of MHW area at the end of May, by an area of roughly 30% of the western Mediterranean, corresponds to overestimations of wind speed by up to 1 m/s and temperature anomalies roughly 1°C weaker than observed. However, the overestimation of activity in July and August, found to be linked to overestimations of SST, does not correspond to overestimations of T2M, implying that other phenomena are not well represented. It should be noted that the use of area-averaged atmospheric variables may hide sub-regional scale processes which impact the MHW location and intensity.

**4. Discussion & Summary**

The MHW of summer 2022 in the Mediterranean Sea was record-breaking, eclipsing 2003 in terms of basin-wide activity (defined as the integral of intensity, duration and area). Other contributions to the Ocean State Report 8 also define the MHW of 2022 as a record-breaking event, using other definitions (e.g. local SST records). Here, we provide a basin-wide view of the MHW conditions. The Copernicus Mediterranean Physical forecasting system was used to track this event, serving as the first validation of MHW prediction for this system. Forecasts captured the full life cycle of the MHWs several days in advance: onset (mid-May) in the Western part of the basin; spread into the Adriatic and Ionian Sea; sporadic local-scale occurrences in the Levantine Basin; persistence of peak conditions throughout July and August; breaks in MHW persistence and abrupt changes in local occurrence; and the gradual decay (September). The forecasts also identified regions shielded from MHWs e.g. during cooling in the Aegean Seas. Subseasonal forecasts do not yet demonstrate the capacity to predict MHW response to weather patterns (Benthuysen et al., 2018), but this study confirms that short-term forecasts, at least in the Mediterranean Sea, can fill this gap.

A full analysis of potential drivers and attribution of forecast skill to certain processes was outside the scope of this study, but the dependence on accurate atmospheric forcings (here provided by ECMWF forecasts) has been shown to be crucial for accurate forecasts of the 2022 event. Unlike the other extreme events, the common drivers of MHWs in the Mediterranean are yet to be identified. The MHW of 2022, as well as the concurrent and record-breaking atmospheric heatwave which occurred over western Europe, appears to be linked to the northward extension of the subtropical ridge (Barriopedro et al., 2023), while model studies have suggested that mid-latitude MHWs in summer typically arise from reduced ocean heat loss to the atmosphere and reduced vertical diffusion (Vogt et al, 2022). Here, the decay in skill of MHW forecasts match the decay in skill of key atmospheric conditions (T2M and wind speed). Erroneous forecast of early summer, in the Western Mediterranean Sea, are explained by inaccurate forecasts of these atmospheric conditions. Peak summer conditions, such as the overestimation of MHW activity, are not yet understood, meaning further studies of short-term forecasting of MHWs are necessary,.

The time scale of forecasting determines the information that can be provided and the type of response to that information. Here, we make the case for using short-term forecasting in MHW tracking tools and studies. Seasonal forecasting informs management decisions and contingency plans, while subseasonal forecasting can update these plans (White et al, 2017). Short-term forecasting, on the other hand, can then be used to determine the precise timings of events and instruct users on when to implement urgent response actions. Longer-term forecasts are typically global in scale and have a relatively low model resolution, while short-term forecasting centres, benefitting from the reduced time scale, can put more computational power towards regional-scale forecasting at a finer scale more relevant to stakeholders. In principle, for MHWs, this means the following: seasonal forecasts forewarn of extreme summer temperatures (e.g. seasonal averages above the 90th percentile, identification of ocean basins affected); sub-seasonal forecasts then update this to forewarn of MHW occurrence (e.g. daily temperatures persisting above 90th percentile, greater detail on geographic spread); finally, short-term forecasts can provide key details such as the start date, onset rate and breaks in occurrence on a local-to-regional scale. Currently, more effort is being placed on seasonal forecasting of MHWs (Liu et al., 2018; Jacox et al., 2022). With the level of accuracy for local-scale MHW indicators shown here, such tools should be complemented with daily, short-term updates.

In particular, we found that the MHW occurrence in the Ligurian Sea and Gulf of Taranto, two regions of economic and ecological importance, was also reliably forecast. There is, though, a need to include subsurface temperatures or heat content to report MHWs occurring at depth (Dayan et al., 2023; McAdam et al., 2023). For example, caged fish have been observed to avoid the top of cages when surface temperatures increase (Gamperl et al., 2021), meaning truly stakeholder-relevant tracking tools need a 3D view. The near-real-time analysis, as well as the forecast system, provides 3D temperatures and can track subsurface propagation of MHWs (unlike satellite observations). The MHW record in the analysis aligns exceptionally well with satellite observations for the two target regions shown (Fig. 3), suggesting a high level of accuracy (the same is found for the basin-wide MHW activity; not shown). However, a subsurface validation with in-situ data should be performed in the near-future, before using the analysis and forecast to track subsurface MHWs.

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

**Code Availability**
Codes used to analyse data and produce figures in this study are available at https://github.com/RJMcAdam.
**Author Contributions**
R.M., G.B., S.M. and E.C. conceived the study. R.M. and G.B. performed the analysis and prepared the figures. R.M wrote
the manuscript. G.B., S.M., E. C contributed to the interpretation of the results and to the paper writing. R.M., G.B., S.M. and
E.C reviewed the manuscript.
**Competing Interests**
The authors no competing interests.

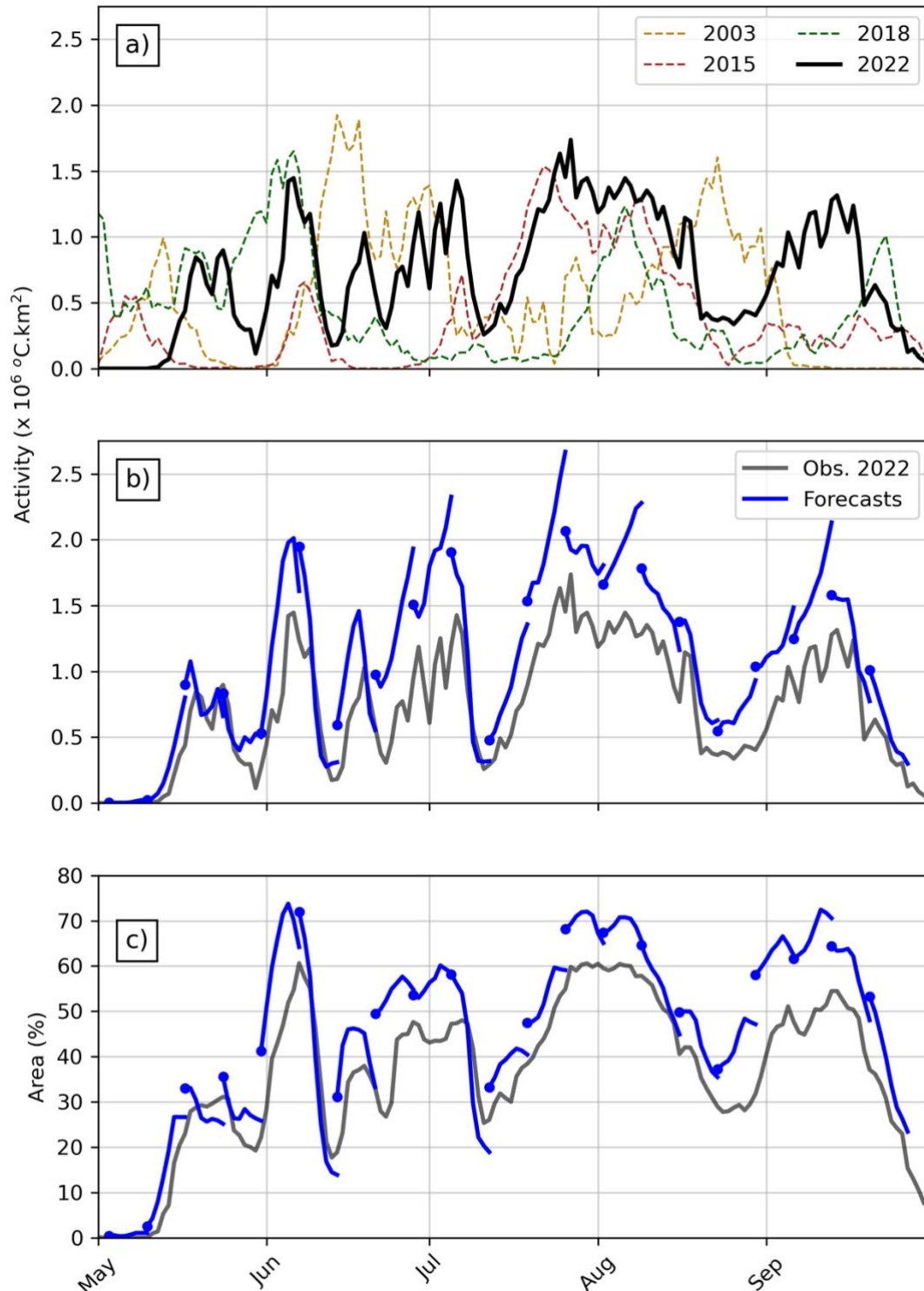


**Figure 1: MHW activity across the Mediterranean Sea.** (a) MHW activity defined by reprocessed satellite observations for 2022 and the
three previous record years according to Simon et al, (2022). (b) Comparison between satellite observations and forecasts of 2022 MHW
activity. (c) Area of Mediterranean Sea experiencing a MHW (as a percentage of total basin area). Activity is defined as the sum of the
intensity over the area undergoing a MHW. Shown here are the first 8 days of forecasts initiated on Tuesdays. Forecast start dates are
shown by the blue dots.

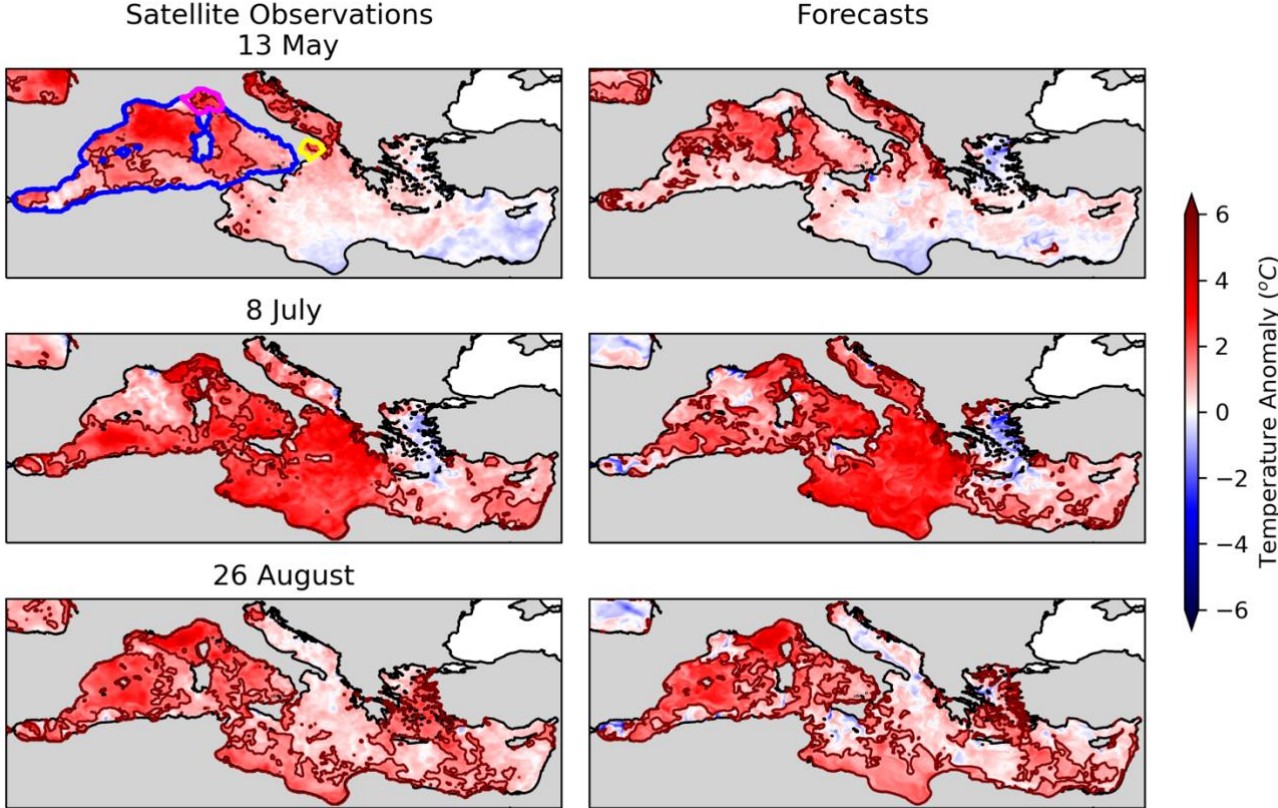


**Figure 2: Snapshots of SST anomalies and MHW occurrence during the different stages of the 2022 MHW.** Left: reprocessed satellite
observations. Right: forecasts with a lead time of 4 days. Areas in which SST is above the 90th-percentile threshold is indicated by the dark
red contour. The 13th May highlights the MHW onset, the 9th July highlights the peak activity, and the 26th August highlights the (first)
decay. Regions used in Figures 3 & 4 are highlighted: Ligurian Sea (magenta), Gulf of Taranto (yellow) and the Western Mediterranean
(blue).

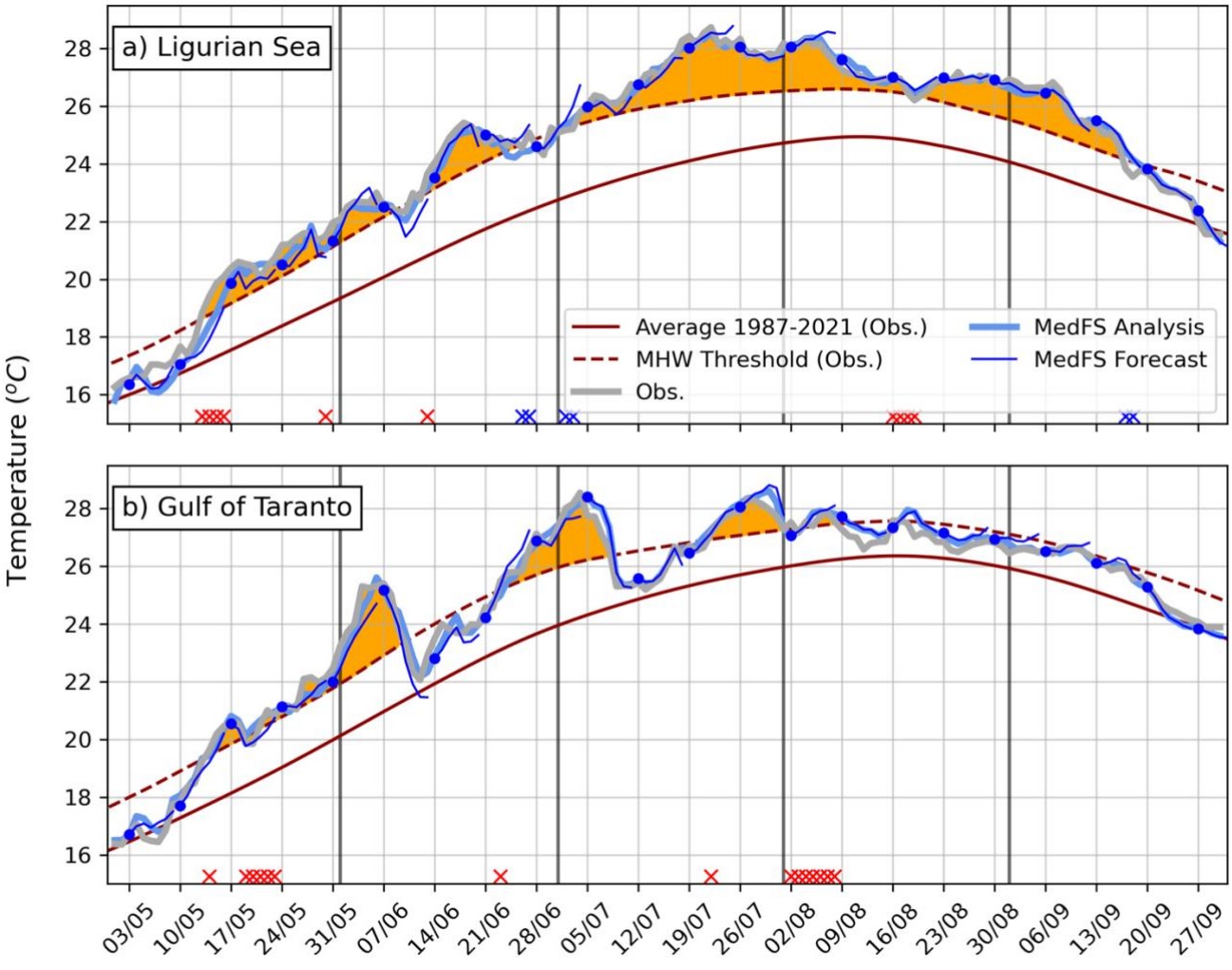


**Figure 3: Time series of SST and MHW occurrence in summer 2022**. Orange (yellow) shading highlights MHW (MHS) occurrence in
reprocessed satellite observations. Forecast start dates are shown by the blue dots. Definitions of the Ligurian Sea (a) and Gulf of Taranto
(b). Note that the climatology lines (red) do not correspond to the satellite data, not to the model output (analysis and forecasts). Crosses
correspond to misses (red) and false alarms (blue) in the forecast output.

482

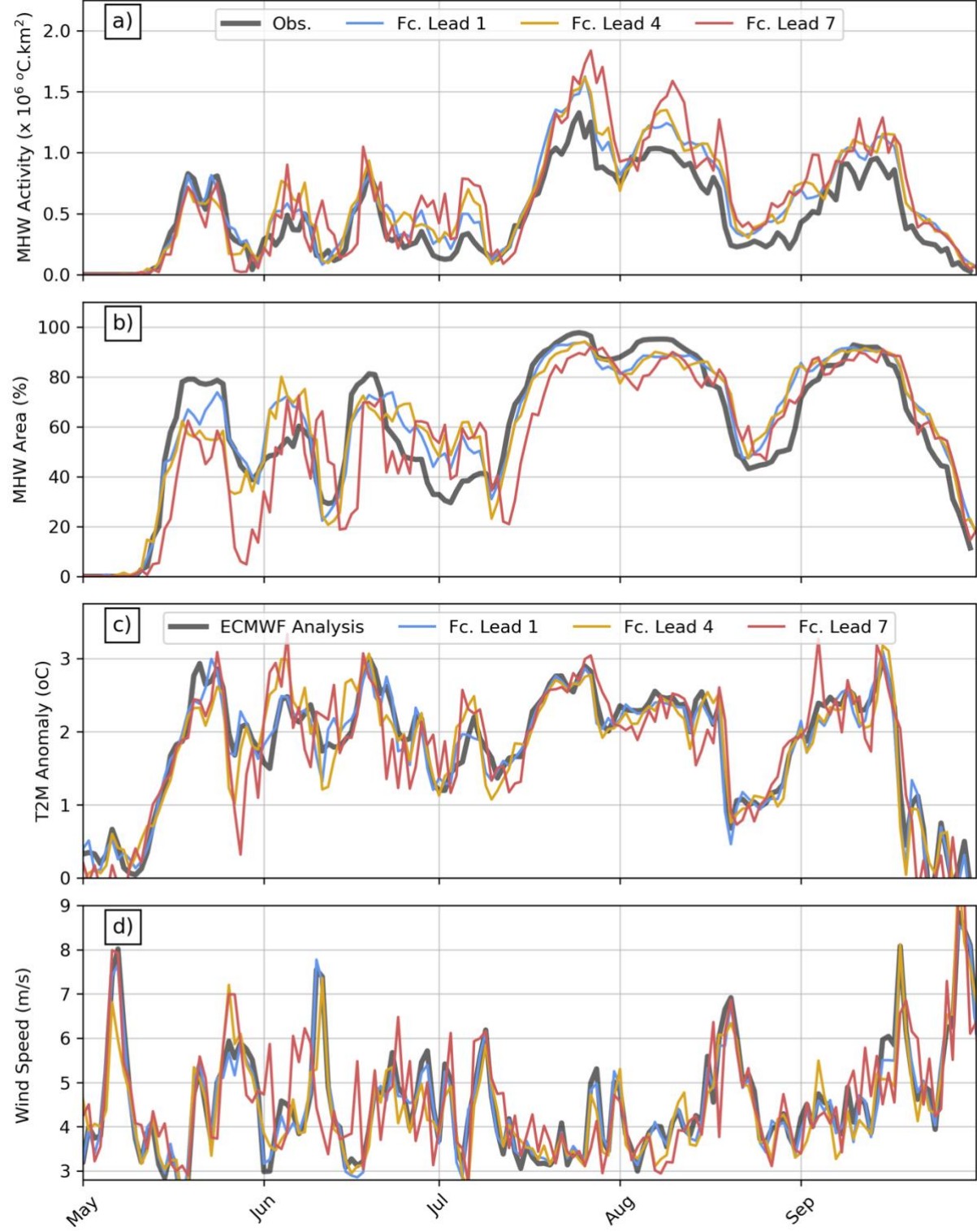

483

**Figure 4: Effect of lead time and atmospheric forcings on forecasts of MHW activity and area**. Comparison between reprocessed satellite observations and forecasts of 2022 MHW activity (a) and area (b). Each forecast time series corresponds to a different lead time (I.e. how many days in advance the forecast was made). Forecasts of MHW activity was calculated for forecasts initiated every day; the lead time from each forecast was extracted to construct the time series. Area-averaged 2m temperature anomaly (c) and wind speed (d) from the ECMWF analysis and forecasts used to force the MedFS system. All time series correspond to the Western Mediterranean Sea (Fig. 2).

Table 1: Root-Mean-Square Differences of forecasts of summer 2022 MHW activity and atmospheric conditions (Fig, 4). Values in parenthesis are RMSD values normalised by standard deviation over the summer. Differences in MHW activity and area are relative to reprocessed satellite observations, while differences jn T2M anomaly and windspeed are relative to ECMWF analysis. Each column corresponds to a different lead time.

| RMSD (Normalised) | Lead: 1 day | Lead: 4 days | Lead: 7 days |
|---|---|---|---|
| MHW Activity | 0.16 (0.48) | 0.20 (0.59) | 0.28 (0.82) |
| MHW Area | 8.88 (0.33) | 11.65 (0.43) | 16.50 (0.61) |
| T2M Anomaly | 0.18 (0.21) | 0.31 (0.38) | 0.52 (0.62) |
| Wind Speed | 0.22 (0.18) | 0.52 (0.42) | 0.94 (0.76) |