# Peer review of "Forecasting the Mediterranean Sea Marine Heatwave of summer 2022"

_State of the Planet, 2023_

## Referee Comment (RC1)

**Forecasting the Mediterranean Sea Marine Heatwave of summer 2022**

**General review**

Marine heatwaves are attracting increasing interest from the research community, but their prediction lacks sufficient attention. This manuscript helps to fill this gap by analysing and extreme marine summer in the Mediterranean. The authors show the skill of a short-term forecasting system at basin-wide and regional scales. Results show the adequacy of short-term forecasting in the Mediterranean with fairly good accuracy.

The topic of the paper is interesting and the approach to it is correct, but it suffers from a certain lack of consistency in justifying the results and conclusions. I am sure that the authors have sufficient data and results to make their conclusions more robust, as the wording is sometimes vague. I think it is more a problem of how to explain the work done than of the work itself, which I think is very relevant.

Through the text you are using different activity definitions. Be consistent throughout the manuscript and use a single definition if possible. If not, please make clear which one you use any time you refer to it. Suggestion, change the name of your definition to "cumulative activity", "basin activity" or similar. This is one of my main concerns as it can be confusing for the reader.

"Discussion and summary" look more like a review paper than the discussion of your results. Consider moving some references to the introduction section or extend the section by adding more discussion of your own results.

Therefore, my recommendation is to accept this work with minor revisions aimed at better clarifying the work developed by the authors. Please, see the comments below.

**Other comments**

Line 24: *"Zi et al., 2020"* do you mean "Li et al., 2020"?

Please check "Benthuysen et al" as there is a typo across the text.

Lines 79-80: *"we define activity as the sum of the intensity over the area undergoing a MHW in the Mediterranean Basin."* But in figure 1 activity is *"the daily product of area and intensity"*. Which is the actual activity metric used in figure 1? Have you used your own definition in the rest of the paper?

Line 94: *"and conditions remained above a third of the basin area until the decay at the end of September"* Above what? Do you mean about SST threshold? Do you mean MHW conditions were present on more than a third of the basin?

Line 96, 103, 105, 155, 156, 166: Please check typo when writing *"°C".*

Lines 104-105. The persistence of a minimum activity (greater than 0) is not enough to state that 2022 holds a new record for MHW activity. Please, add more justification for this statement (higher mean, higher max, greater area…) that explains the record.

Line 150: Why do you choose the Ligurian Sea and Gulf of Taranto? Why not the Alboran Sea if maritime activity is the selection criteria? Why not other areas of interest? You could choose many other criteria (oceanic circulation, upwelling areas, heavy precipitation prone coastal areas, high biodiversity areas, highest MHW intensity in 2022,…). Please, better explain why

using maritime activity. Later in the same paragraph biological importance of these two areas is explained.

Lines 156-157 "*This activity is indicative of the conditions experienced by the rest of the western part of the Mediterranean basin*". How do you support this statement? Not all the western basin experienced MHW and its intensity showed noticeable variability. Do you mean that the Ligurian Sea is a proxy for the whole WMED?

Line 159: Please add a value (percentage?) for *"the vast majority of days"*.

Line 170: *"upon visual inspection the forecast temperature was very similar to the observed"*. Could you add some mean bias or other accuracy measure?

Lines 171-176: Is the analysed accuracy basin-wide? Or for the two previously mentioned areas? You should extend this paragraph as it does not provide a clear idea of the forecast accuracy.

Line 180: Another definition of activity. Please check out consistency of activity definition across the text.

Lines 180-181: *"Other contributions to this report also define 180 the MHW of 2022 as a record-breaking event, using various other definitions."* Please, better explain this sentence. It does not make sense in its actual form.

Line 188: Please use an acronym or the full name for the Copernicus Med Phys forecasting system instead of *"the system"*.

Lines 188—192: Yours and other author results show the CMPF system is capable to forecast a range of extreme events, but this is not a conclusion of your work. This would fit better in the introduction or methods section to justify the use of CMPF data.

**Figures**

Figure 1b-c. Please, add the grey line to the legend. I assume it is satellite data.

Figure 3: Please, change colour for Near-Real-Time Obs. And MEDFS Analysis. They are too similar and confusing right now.

Figure 4: Please, add the satellite to the legend and caption (thick grey?)

General recommendation: Some lines on the figures look hand-drawn and some of them overlap the other so it is difficult to understand. Please, try to improve figures readability.

---

## Author Comment (AC1)

This short paper investigates the predictability of the Marine Heatwave that occurred in the Mediterranean Sea in summer 2022. Using short-term (10 days) forecasts from Copernicus Marine Services, it examines if this system is able to successfully forecast the onset, evolution, peaks of this extreme event, first at basin scale, and then at two specific locations of economic importance. The main message is that short-term forecasts exhibit some skill in predicting details of the MHW evolution a few days in advance, which is of importance for guiding management.

This paper addresses an important challenge. Marine Heatwaves have recently received much attention because of their damaging impacts on marine ecosystems; several events have struck the Mediterranean Sea in recent years. It is of great importance to investigate their potential predictability at different lead times. While several papers focused on seasonal forecasts, this study focus on short-term (10 days), which has received little attention so far.

The paper is well written; the approach (first at basin scale, then at local scale) seems relevant to the objectives, and the results are worth being published.

We thank the reviewer for their careful reading and instructive comments which have helped improve the quality of the manuscript.

Yet, I have two major concerns:

- the analyses presented are mostly qualitative, and would deserve more quantification. Sentences like " the area of MHW conditions was also well predicted" (line 116) or "MHW occurrence [...] correctly forecast" (line 137) or "upon visual inspection the forecast temperature was very similar to the observed" are subjective. What is a "good" forecast? How can it be defined? What are the acceptable errors? Spatial correlations, quantification of the differences, or other metrics to better quantify these are needed. Figure 4 is not fully discussed, yet the results do not seem very encouraging for 7-days lead time. Are the forecasts "good" for this lead-time?

  These sentences have been backed up with quantification or reworded. For example, Lines 126-132 now refers to the timing of MHWs being correctly forecast. "Accurate" is reserved for statements backed up by quantification. We have added Table 1 to show

Root-Mean-Square-Differences (RMSD) of the forecasts shown in Figure 4, with values normalised by the standard deviation to show where errors exceed the natural variability. Figure 4 now includes forecasts of MHW area, which allows us to state whether overestimates arise from errors in area or intensity.

Given that these indicators (activity and area) have not been studied in short-term forecasts before, it is not possible to compare. The measures of normalised RMSD show that these indicators are predicted as accurately as atmospheric variables (values <1).

Table 1: Root-Mean-Square Differences of forecasts of summer 2022 MHW activity and atmospheric conditions (Fig, 4). Values in parenthesis are RMSD values normalised by standard deviation over the summer. Differences in MHW activity and area are relative to reprocessed satellite observations, while differences jn T2M anomaly and windspeed are relative to ECMWF analysis. Each column corresponds to a different lead time.

| RMSD (Normalised) | Lead: 1 day | Lead: 4 days | Lead: 7 days |
|---|---|---|---|
| MHW Activity | 0.16 (0.48) | 0.20 (0.59) | 0.28 (0.82) |
| MHW Area | 8.88 (0.33) | 11.65 (0.43) | 16.50 (0.61) |
| T2M Anomaly | 0.18 (0.21) | 0.31 (0.38) | 0.52 (0.62) |
| Wind Speed | 0.22 (0.18) | 0.52 (0.42) | 0.94 (0.76) |

[Figure]

**Figure 4: Effect of lead time and atmospheric forcings on forecasts of MHW activity and area**. Comparison between reprocessed satellite observations and forecasts of 2022 MHW activity (a) and area (b). Each forecast time series corresponds to a different lead time (I.e. how many days in advance the forecast was made). Forecasts of MHW activity was calculated for forecasts initiated every day; the lead time from each forecast was extracted to construct the time series. Area-averaged 2m temperature anomaly (c) and wind speed (d) from the ECMWF analysis and forecasts used to force the MedFS system. All time series correspond to the Western Mediterranean Sea (Fig. 2).

- The analyses remain descriptive, without analyses on the processes that can lead to improved or degraded forecasting capability. Although I understand that a full characterization of the 2022 MHW drivers with a heat budget may be beyond the scope of this paper, it would be useful to provide more hints on the underlying processes, so that the scope of this study would not be restricted to this particular event but would provide information useful for other upcoming events.

  To Figure 4 we have added forecasts from the ECMWF system, used to forecast the MedFS system, of T2M anomaly and wind speed. We highlight that Figure 4 now corresponds to the Western Mediterranean, not the full basin, to facilitate a study of drivers. We believe that area averages of T2m and wind speed over the whole basin will lead to masking signals; the Western Med is chosen as a compromise between studying drivers and representing the extent of the MHW.

In particular, one interesting result is the contrast between the two periods in early summer and late summer (May-mid-June versus July-August-September) as seen in Figure 1b. The operational system exhibit better forecasting skill during the early period, whereas it fails to predict the sharp increase in activity (intensity) in mid-July, early August, early September. Looking at the three specific periods to examine which processes in the real world led to MHW decrease instead of intensification, would help to understand these false alarms. Line 134, the authors suggest as potential candidates "cloud cover or winds". Would it be possible to investigate this?

As mentioned above, we include an analysis of forcing forecasts of T2m and wind speed. Line 252-262: *"The decreases in skill with lead time can partly be explained by the decrease in skill of the ECMWF atmospheric forecasts used to force the Mediterranean Sea forecasts system. T2M and wind speed correlate strongly and significantly with the MHW activity (correlation values of 0.89 and 0.50 with the ECWMF analysis respectively), evidencing their role in MHW formation. Errors of forecasts of T2M and wind speed grow with time but do not exceed natural variability at lead time 7 (Table 1). In the first half of the summer, forecasts at lead time 7 of both T2M and wind speed are frequently out of phase with the observed changes. In fact, the underestimations of MHW area in this period occur simultaneously with underestimations of T2M. For example, the underestimation of MHW area at the end of May, by an area of roughly 30% of the western Mediterranean, corresponds to*

*overestimations of wind speed by up to 1 m/s and temperature anomalies roughly 1°C weaker than observed. However, the overestimation of activity in July and August, found to be linked to overestimations of SST, does not correspond to overestimations of T2M, implying that other phenomena are not well represented. It should be noted that the use of area-averaged atmospheric variables may hide sub-regional scale processes which impact the MHW location and intensity."*

Finally, some details are missing. The area chosen for the definition of the Ligurian Sea and Gulf of Taranto (shown in Figure 3) are not given. In Figure 4, it should be stated that the Figure is the basin average.

We thank the reviewer for spotting these missing details. The areas have been added to Figure 2. Figure 4 now corresponds to MHW activity (integrated over the western Mediterranean Sea) and MHW area. The atmospheric variables in Fig. 4c and 4d are area-averaged.

Minor comments:

-line 24: McAdam et al. 2023 reference is missing

We have now added this reference.

-line 34: I disagree with the statement that 5 days of longer are harmful to marine life. This is completely species-dependent. The Hobday et al. (2016) definition is just a practical one, without reference to ecosystems impacts.

We have updated this sentence to highlight the reviewers point: *"The definition of MHWs assumes persistent conditions are harmful to marine life if the duration is 5 days or longer (Hobday et al., 2016), although this number is quite arbitrary and in principle should be species-dependent. "*

- line 74: no trend is removed. This choice may be usefully justified, given the recent debates from the community about removing or not a trend for MHW detection

We have added to description of how MHWs are calculated: Line 98: *"Although there are benefits of detrending SST prior to detecting MHWs (Amaya et al., 2023), we chose not to detrend in order to present the true values of temperature because they are of more relevance to species impacts (e.g. Galli et al., 2017)."*

---

## Author Comment (AC2)

Marine heatwaves are attracting increasing interest from the research community, but their prediction lacks sufficient attention. This manuscript helps to fill this gap by analysing and extreme marine summer in the Mediterranean. The authors show the skill of a short-term forecasting system at basin-wide and regional scales. Results show the adequacy of short-term forecasting in the Mediterranean with fairly good accuracy.

The topic of the paper is interesting and the approach to it is correct, but it suffers from a certain lack of consistency in justifying the results and conclusions. I am sure that the authors have sufficient data and results to make their conclusions more robust, as the wording is sometimes vague. I think it is more a problem of how to explain the work done than of the work itself, which I think is very relevant.

We thank the reviewer for their careful reading and instructive comments which have helped improve the quality of the manuscript. We have made attempts to improve wording and messaging. Please see below for responses to the specific comments.

Through the text you are using different activity definitions. Be consistent throughout the manuscript and use a single definition if possible. If not, please make clear which one you use any time you refer to it. Suggestion, change the name of your definition to "cumulative activity", "basin activity" or similar. This is one of my main concerns as it can be confusing for the reader.

We use only one definition of activity, as was previously defined in Line 104: "Here, in order to study basin-wide spread at daily resolution, we define activity as the sum of the intensity over the area undergoing a MHW in the Mediterranean Basin". We have corrected the caption of Figure 1, which we believe was the source of confusion. We have changed "activity" to "occurrence" where necessary.

"Discussion and summary" look more like a review paper than the discussion of your results. Consider moving some references to the introduction section or extend the section by adding more discussion of your own results.

The discussion section has now been updated. The discussion of forecasting other extreme events has been moved to the introduction.

Therefore, my recommendation is to accept this work with minor revisions aimed at better clarifying the work developed by the authors. Please, see the comments below.

**Other comments**

Line 24: *"Zi et al., 2020"* do you mean "Li et al., 2020"?

Yes, thank you for finding this.

Please check "Benthuysen et al" as there is a typo across the text.

Thank you for finding this, the name has been corrected across the text.

Lines 79-80: *"we define activity as the sum of the intensity over the area undergoing a MHW in the Mediterranean Basin."* But in figure 1 activity is *"the daily product of area and intensity"*. Which is the actual activity metric used in figure 1? Have you used your own definition in the rest of the paper?

We apologise for the confusion. We use only one definition of activity, as was previously defined in Line 104: "Here, in order to study basin-wide spread at daily resolution, we define activity as the sum of the intensity over the area undergoing a MHW in the Mediterranean Basin". We have corrected the caption of Figure 1. We have changed "activity" to "occurrence" where necessary.

Line 94: *"and conditions remained above a third of the basin area until the decay at the end of September"* Above what? Do you mean about SST threshold? Do you mean MHW conditions were present on more than a third of the basin?

We have corrected this sentence to make it clear we are referring to MHW area being above a third of the total basin area.

Line 96, 103, 105, 155, 156, 166: Please check typo when writing *"°C".*

Thank you for spotting this formatting issue; we have corrected it across the text.

Lines 104-105. The persistence of a minimum activity (greater than 0) is not enough to state that 2022 holds a new record for MHW activity. Please, add more justification for this statement (higher mean, higher max, greater area...) that explains the record.

Unfortunately we do not understand the confusion of the reviewer here. A justification was already included in Lines 128-134.

Line 150: Why do you choose the Ligurian Sea and Gulf of Taranto? Why not the Alboran Sea if maritime activity is the selection criteria? Why not other areas of interest? You could choose many other criteria (oceanic circulation, upwelling areas, heavy precipitation prone coastal areas, high biodiversity areas, highest MHW intensity in 2022,...). Please, better explain why using maritime activity. Later in the same paragraph biological importance of these two areas is explained.

We have already justified why these two regions are important. Lines 182-231. The reviewer is right to point out the economic and ecologic importance of other areas, although that argument can be expanded to most of the Mediterranean Sea. Unfortunately there is not enough space for more case studies in the Ocean State Report.

Lines 156-157 "*This activity is indicative of the conditions experienced by the rest of the western part of the Mediterranean basin*". How do you support this statement? Not all the western basin experienced MHW and its intensity showed noticeable variability. Do you mean that the Ligurian Sea is a proxy for the whole WMED?

This sentence has been removed.

Line 159: Please add a value (percentage?) for *"the vast majority of days"*.

We have added the value of 89%.

Line 170: *"upon visual inspection the forecast temperature was very similar to the observed"*.

Could you add some mean bias or other accuracy measure?

We have now added a Table of Root-Mean-Square differences (Table 1), with normalised values to indicate the magnitude of errors w.r.t natural variability.

Table 1: Root-Mean-Square Differences of forecasts of summer 2022 MHW activity and atmospheric conditions (Fig, 4). Values in parenthesis are RMSD values normalised by standard deviation over the summer. Differences in MHW activity and area are relative to reprocessed satellite observations, while differences jn T2M anomaly and windspeed are relative to ECMWF analysis. Each column corresponds to a different lead time.

| RMSD (Normalised) | Lead: 1 day | Lead: 4 days | Lead: 7 days |
|---|---|---|---|
| MHW Activity | 0.16 (0.48) | 0.20 (0.59) | 0.28 (0.82) |
| MHW Area | 8.88 (0.33) | 11.65 (0.43) | 16.50 (0.61) |
| T2M Anomaly | 0.18 (0.21) | 0.31 (0.38) | 0.52 (0.62) |
| Wind Speed | 0.22 (0.18) | 0.52 (0.42) | 0.94 (0.76) |

Lines 171-176: Is the analysed accuracy basin-wide? Or for the two previously mentioned areas? You should extend this paragraph as it does not provide a clear idea of the forecast accuracy.

We have added specific values of accuracy to the two regions (e.g. Line 215), as well as to the Western Mediterranean Sea (Table 1.)

Line 180: Another definition of activity. Please check out consistency of activity definition across the text.

We have changed "activity" to "occurrence" where necessary.

Lines 180-181: *"Other contributions to this report also define 180 the MHW of 2022 as a record-breaking event, using various other definitions."* Please, better explain this sentence. It does not make sense in its actual form.

We have modified this phrase: *"Other contributions to the Ocean State Report 8 also define the MHW of 2022 as a record-breaking event, using other definitions (e.g. local SST records)."*

Line 188: Please use an acronym or the full name for the Copernicus Med Phys forecasting system instead of *"the system".*

We have used "MedFS" where necessary.

Lines 188—192: Yours and other author results show the CMPF system is capable to forecast a range of extreme events, but this is not a conclusion of your work. This would fit better in the introduction or methods section to justify the use of CMPF data.

This point has now been moved to the introduction.

**Figures**

Figure 1b-c. Please, add the grey line to the legend. I assume it is satellite data.

We have added the grey line which represents reprocessed satellite data.

Figure 3: Please, change colour for Near-Real-Time Obs. And MEDFS Analysis. They are too similar and confusing right now.

We have changed colours.

Figure 4: Please, add the satellite to the legend and caption (thick grey?)

This has been added to the legend and caption.

General recommendation: Some lines on the figures look hand-drawn and some of them overlap the other so it is difficult to understand. Please, try to improve figures readability.

We have attempted to improve the clarity of the figures, especially Figures 3 and 4 which contain lots of information.